# Dietary Intake and Diet Quality of Adult Survivors of Childhood Cancer and the General Population: Results from the SCCSS-Nutrition Study

**DOI:** 10.3390/nu13061767

**Published:** 2021-05-22

**Authors:** Fabiën N. Belle, Angeline Chatelan, Rahel Kasteler, Luzius Mader, Idris Guessous, Maja Beck-Popovic, Marc Ansari, Claudia E. Kuehni, Murielle Bochud

**Affiliations:** 1Childhood Cancer Research Group, Institute of Social and Preventive Medicine (ISPM), University of Bern, 3012 Bern, Switzerland; rahel.kasteler@ispm.unibe.ch (R.K.); luzius.mader@ispm.unibe.ch (L.M.); claudia.kuehni@ispm.unibe.ch (C.E.K.); 2Center for Primary Care and Public Health (Unisanté), University of Lausanne, 1010 Lausanne, Switzerland; angeline.chatelan@unisante.ch (A.C.); murielle.bochud@unisante.ch (M.B.); 3Children’s University Hospital of Bern, University of Bern, 3010 Bern, Switzerland; 4Division and Department of Primary Care Medicine, Geneva University Hospital HUG, 1205 Geneva, Switzerland; Idris.Guessous@hcuge.ch; 5Pediatric Hematology-Oncology Unit, Lausanne University Hospital CHUV, 1010 Lausanne, Switzerland; Maja.Beck-Popovic@chuv.ch; 6Pediatric Onco-Hematology Unit, Geneva University Hospital HUG, 1205 Geneva, Switzerland; marc.ansari@hcuge.ch; 7Cansearch Research Laboratory, Geneva Medical School, 1205 Geneva, Switzerland

**Keywords:** dietary intake, AHEI, food frequency questionnaire, childhood cancer survivors, Swiss Childhood Cancer Registry, Europe

## Abstract

Childhood cancer survivors (CCSs) are at increased risk of developing chronic health conditions. This may potentially be reduced by a balanced diet. We aimed to compare dietary intake and diet quality using the Alternative Healthy Eating Index (AHEI) of adult CCSs and the general Swiss population. A food frequency questionnaire (FFQ) was completed by CCSs with a median age of 34 (IQR: 29–40) years. We compared dietary intake of 775 CCSs to two population-based cohorts who completed the same FFQ: 1276 CoLaus and 2529 Bus Santé study participants. CCSs consumed particular inadequate amounts of fiber and excessive amounts of sodium and saturated fat. Dietary intake was similar in CCSs and the general population. The mean AHEI was low with 49.8 in CCSs (men: 47.7, women: 51.9), 52.3 in CoLaus (men: 50.2, women: 54.0), and 53.7 in Bus Santé (men: 51.8, women: 54.4) out of a maximum score of 110. The AHEI scores for fish, fruit, vegetables, and alcohol were worse in CCSs than in the general population, whereas the score for sugar-sweetened beverages was better (all *p* < 0.001). Diet quality at follow-up did not differ between clinical characteristics of CCSs. Long-term CCSs and the general population have poor dietary intake and quality in Switzerland, which suggests similar population-based interventions for everyone.

## 1. Introduction

A healthy diet is a modifiable factor that can affect the development of chronic diseases such as Type 2 diabetes, metabolic syndrome, and cardiovascular disease (CVD) [1,2,3]. To prevent these health problems, it is recommended to consume a diet that is rich in fruit, vegetables, fiber, and complex carbohydrates, but low in saturated and trans-fatty acids, and to limit alcohol consumption [1,2,3]. Accumulating research among childhood cancer survivors (CCSs) shows that the burden of chronic diseases secondary to childhood cancer or treatment can be reduced with dietary adaptation, weight management, and physical activity [4,5,6,7]. The importance of a healthy lifestyle was confirmed in a study of 1598 CCSs 26 years postdiagnosis [6], which found that an unhealthy lifestyle based on anthropometric, FFQ, and physical activity data according to World Cancer Research Fund/American Institute for Cancer Research recommendations was associated with higher risk of metabolic syndrome. This was confirmed in the PETALE study including 241 CCSs; poor adherence to dietary recommendations was adversely associated with cardiometabolic health [8]. A healthy diet is therefore particularly relevant for CCSs who are at increased risk of developing chronic diseases, yet few studies have evaluated the intake or quality of the diet of CCSs [9,10].

Poor adherence to the 2010 Dietary Guidelines for Americans and poor diet quality based on the Healthy Eating Index-2010 (HEI-2010, 58% of the maximum score) were found in 2570 adult long-term CCSs in the St. Jude Lifetime cohort [11]. Diet quality was poorer in CCSs that had been diagnosed young and were treated with abdominal radiation therapy. Similar results have been observed in smaller studies in the United States (US) and Canada, reporting that CCSs adhered poorly to dietary recommendations. Mean diet quality in these studies ranged from 33% to 60% of the maximum score [5,6,8,12,13,14]. A Chinese study including 181 CCSs reported less soft drink, sugar, and fast-food consumption compared to Western studies. However, 55% of the CCSs reported picky eating, of whom 58% refused to eat vegetables [15].

Studies of lifestyle and eating habits among CCS outside the US are few. They have small sample sizes, short follow-up times, or focus on specific cancer diagnoses [8,12,16,17]. Most previous studies lack comparison groups to compare dietary intake and quality between CCSs and the general population. To cover these gaps, we aimed at analyzing data from the Swiss Childhood Cancer Survivor Study (SCCSS)-Nutrition study to assess dietary intake and diet quality of CCSs in comparison with the general Swiss adult population. The comparison with random population samples is important to see whether dietary intake are influenced by childhood cancer and/or its treatment. The second aim of the study was to explore whether clinical characteristics in CCSs at baseline were associated with diet quality at follow-up.

## 2. Materials and Methods

### 2.1. Study Populations

#### 2.1.1. The Swiss Childhood Cancer Survivor Study

The SCCSS is a population-based, long-term follow-up study of all childhood cancer patients registered in the Swiss Childhood Cancer Registry (SCCR) who were diagnosed in Switzerland with leukemia, lymphoma, central nervous system (CNS) tumors, malignant solid tumors, or Langerhans cell histiocytosis; who were under the age of 21 years at time of diagnosis; who survived ≥5 years after initial diagnosis of cancer; and who were alive at the time of the study [18,19,20]. Ethical approval of the SCCR and the SCCSS was granted by the Ethics Committee of the Canton of Bern (KEK-BE: 166/2014); the SCCSS is registered at clinicaltrials.gov (NCT03297034).

CCSs were eligible to participate in the SCCSS-Nutrition study if they had childhood cancer diagnosed between 1976 and 2005, completed a baseline SCCSS questionnaire between 2007 and 2013 [18], and were 21 years of age or older at the time of follow-up survey in 2017. We sent a follow-up questionnaire that included a food frequency questionnaire (FFQ) to all CCSs who were enrolled in SCCSS-Nutrition. Detailed information on the SCCSS-Nutrition study design can be found in a prior publication [21].

#### 2.1.2. Comparison Groups

We used two population-based samples of the Swiss adult population represented by data from the Bus Santé survey and the Cohorte Lausannoise (CoLaus) study which assessed dietary intake similarly as in CCSs. Bus Santé is a cross-sectional survey that is ongoing in the canton of Geneva [22]. A representative sample of non-institutionalized men and women aged 20–74 years has been recruited each year since 2012. For this study we included the surveys between 2015 and 2018. Eligible participants are identified with a standardized procedure using a residential list established by the local government [23]. Random sampling in age and sex-specific strata is proportional to the corresponding frequencies in the population. Those who did not respond after three mailings and seven phone calls were replaced using the same selection protocol, but those who declined to participate were not replaced. Participants were not eligible for future recruitments and surveys. Participants received a self-administered questionnaire including a FFQ (described below) to collect data on sociodemographic characteristics, health behaviors, and dietary intake at home before receiving an invitation to a health examination in a clinic or a mobile medical unit. During the examination, trained staff checked the questionnaires for completeness. The study was approved by the Ethics Committee of Canton Geneva (IRB00003116).

The CoLaus study (www.colaus-psycolaus.ch, accessed on date: 25 February 2021) is a prospective cohort study conducted in the city of Lausanne to identify biological and genetic determinants of CVD. From June 2003 until May 2006, a random sample of middle-aged men and women were recruited for the baseline examination. Those who participated in the baseline study were asked to participate in the follow-up study between April 2009 and September 2012 [24]. In the follow-up study, participants were over 40 years old and dietary intake was assessed with the same FFQ that was used in the Bus Santé surveys and the SCCSS-Nutrition study. The study was approved by the institutional Ethics Committee of the University of Lausanne (approvals 16/03 and 33/09). All participants of both comparison groups provided written informed consent.

### 2.2. Measurements 

#### 2.2.1. Dietary Intake

Dietary intake of CCSs and participants in the Bus Santé and CoLaus surveys was assessed using the same self-administered, semiquantitative FFQ including portion sizes [25,26]. The FFQ was originally developed and validated for the French-speaking Swiss adult population [22,25,27,28], (Appendix A). It collects information on consumption frequency and portion sizes of 97 fresh and prepared food items (excluding dietary supplements), organized in 12 different food groups, during the previous four weeks. Consumption frequencies range from never during the last four weeks to two or more times per day, and portions are divided into three sizes equal to, or smaller or larger than a reference size. The reference portions are defined as common household measures representing the median portion size of a previous validation study performed with 24-h dietary recalls [28]. The smaller and larger portion sizes represented the first and fourth quartiles of this distribution. The French Information Center on Food Quality (Centre d’Information sur la Qualité des Aliments) food composition table [29] and the Swiss Food Composition Database of the Federal Food Safety and Veterinary Office [30] were used to convert the food portions into macronutrients and micronutrients. We calculated daily sodium intake using an equation developed specifically for this FFQ for males and females separately [27,31]. This equation is based on calibrations on total salt intake from 24-h urine collections in a validation study that included 100 healthy people.

#### 2.2.2. Sociodemographic, Lifestyle and Clinical Characteristics 

For all CCSs and comparison groups, we collected self-reported data on sex, age at survey, educational level, country of birth, language region in Switzerland, living situation (living alone or with others), physical activity, smoking status, and body mass index (BMI) at survey. Physical activity was assessed differently in CCSs and comparison groups. In the SCCSS we dichotomized physical activity into two groups according the WHO guidelines for physical activity in adults: inactive (lower than 150 min of activity per week); and active (150 min or more of moderate or 75 min of vigorous physical activity, or a combination of moderate and vigorous intense physical activity per week) [32]. In the comparison groups physical activity was assessed with a validated, self-administered physical activity frequency questionnaire (PAFQ) [33] and dichotomized into inactive (first quartile of total weekly physical activity time excluding sleep), or active (more than the first quartile). For all CCSs, we had self-reported information on weight without clothes, and height without shoes, at time of survey. In the two random samples of the general Swiss adult population, body weight and height were measured without shoes, in light indoor clothes. Body weight was measured in kilograms to the nearest 100 g using a calibrated electronic scale (Seca^®^, Hamburg, Germany). Height was measured to either the nearest 5 or 10 mm using a Seca^®^ height gauge. We calculated BMI by dividing weight in kilograms by height in meters squared (kg/m^2^) in all groups. BMI was classified as underweight (<18.5 kg/m^2^), normal (≥18.5 to <25 kg/m^2^), overweight (≥25 to <30 kg/m^2^), or obese (≥30 kg/m^2^) [34]. 

For the CCSs population, we extracted additional clinical information from the SCCR. Cancer diagnosis was classified according to the International Classification of Childhood Cancer, Third Edition [35]. Radiotherapy was classified as any, cranial, chest, total body and/or abdominal, or no radiotherapy. Cranial radiation was considered as present if the survivor had received direct radiation to the brain and/or skull. Chest radiotherapy was defined as direct radiation applied to the chest including total body irradiation, mantlefield irradiation, or irradiation of the thorax, mediastinum, or thoracic spine. Cumulative dosage of radiotherapy was obtained from medical records and categorized according to the Children’s Oncology Group Long-Term Follow-up (COG-LTFU) guidelines. Irradiation was categorized as <18 Gray (Gy) or ≥18 Gy for cranial irradiation; the chest, <30 Gy or ≥30 Gy; and as total body and/or abdominal irradiation versus no radiation [36]. Other treatments were divided into glucocorticoids, anthracyclines, alkylating agents, and hematopoietic stem cell transplantation (HSCT). Glucocorticoid intake, including prednisone and/or dexamethasone, was estimated based on the cancer treatment protocols as described previously [37]. We also retrieved records on relapse during follow-up. 

#### 2.2.3. Statistical Analyses 

We included all CCSs and participants from the general population who were aged 20–50 years at time of survey (2015–2018 for Bus Santé and 2009–2012 for the CoLaus study), who provided reliable dietary intake information (>850 kcal or <4500 kcal per day), and were neither pregnant nor lactating during the survey (information available only in CCSs). For better comparison between CCSs and peers, we standardized comparison groups for sex and age at survey, as previously described [38]. The first step in our analyses was to evaluate whether CCS and their peers met the dietary recommendations for Germany (D), Austria (A), and Switzerland (CH) (DACH) [39]. We compared mean intake to the recommended intake or, when not available, the adequate intake. We calculated mean intake based on age and sex recommendations weighted by the age and sex distribution of the study population. Nutritional goals were set at 100% when the mean intake met the recommended or adequate intake. Total energy intake was calculated including calories from alcohol consumption. To estimate the diet quality of CCSs and peers we used the modified Alternative Healthy Eating Index (AHEI) [40]. The AHEI is composed of 11 food- and nutrient-specific components, including vegetables, fruit, whole grain, sweetened beverage and fruit juice, nuts, seeds, legumes, and tofu, red and processed meat, trans-fat, fish (as a proxy for long-chain n-3 fatty acids), polyunsaturated fatty acids (PUFA), sodium, and alcohol intake [40]. All components of the modified AHEI score range from zero (worst) to ten (best), and the total AHEI score ranges from zero (nonadherence) to 110 (perfect adherence). We compared whether the AHEI score differed between CCSs, Bus Santé and CoLaus data. We exploratory assessed how dietary quality of CCSs differed by cancer diagnosis and treatment with ANCOVA while adjusting for sex, age, and language region. We additionally adjusted treatment exposures for cancer diagnosis. We did not adjust for education level, smoking habits, physical activity, and BMI because these covariates can be affected by cancer diagnosis, treatment exposures, and the occurrence of late effects (i.e., potential intermediates on the causal pathway). We used Stata (version 16, Stata Corporation, Austin, TX, USA) for all analyses.

## 3. Results

### 3.1. Response Rate and Characteristics of the Study Populations

Among 1749 eligible CCSs, we traced and contacted 1599, 919 of whom (57%) returned a questionnaire. We excluded 11 who were pregnant or breastfeeding, 35 who did not report their dietary intake, 71 who had unreliable dietary intake data (<850 kcal or >4500 kcal per day), and a further 27 survivors who were over 50 years old. We thus included 775 CCSs in this study (Appendix A). The participation rates in the Bus Santé survey ranged from 55 to 65% between 2005 and 2017 [23]. Of the 4552 participants in the Bus Santé survey between 2015 and 2018, 1782 were over 50 years old, 88 did not report their dietary intake, and 153 had unreliable dietary intake information. Leaving 2529 for this analysis. Of the 5064 participants in the 2009–2012 follow-up of the CoLaus study, which had a participation rate of 41%, we excluded 3616 because they were over 50 years old, 126 because they had no dietary intake information, and 46 because of unreliable dietary intake, leaving 1276 participants for the analyses.

More CCSs than peers were born in Switzerland since the initial inclusion criterion of the SCCSS stipulated living in Switzerland at time of cancer diagnosis (all *p* < 0.001). CCSs earned a university degree less often (*p* < 0.001) and were less likely to live alone (*p* < 0.001), than those in the Bus Santé study. CCSs were more physical active (*p_Bus Santé_* = 0.002, *p_CoLaus_* < 0.001) and smoked less (all *p* < 0.001) than comparison groups. CCSs had more often a normal BMI than those in the CoLaus study (*p* < 0.001) but less often than those in the Bus Santé study (*p* = 0.015). Characteristics of CCSs and comparison groups are given in Table 1.

Among CCSs, the most common cancers were leukemia, lymphoma, and CNS tumors (Appendix A). Median age at diagnosis was nine years (IQR: 4–14 years) and median time from diagnosis to survey was 26 years (IQR: 20–31 years). Eleven percent had experienced a relapse. CCSs had been treated with glucocorticoids (43%), alkylating agents (41%), anthracyclines (38%), any radiation (37%), and hematopoietic stem cell transplantation (4%).

### 3.2. Dietary Intake and Diet Quality in CCSs and Comparison Groups

Reported total energy intake was low across all populations (Appendix A), in particular in CCSs (1638 kcal/day versus Bus Santé: 1732 kcal/day and CoLaus: 1923 kcal/day). Dietary intakes of CCSs and comparison groups, compared to the dietary DACH recommendations, are represented in Figure 1. Although the differences in percentages of the recommended DACH intake or limit were statistically significant between CCSs and the comparison groups, the absolute differences were small (Appendix A and Figure 1). Nevertheless, CCSs had a substantially lower daily intake of alcohol (5.7 g vs. Bus Santé: 7.9 g, CoLaus: 9.3 g). In CCSs, intakes higher than recommended, considering age and sex distributions, were seen for sodium, phosphorus, saturated fat, cholesterol, protein, and total fat. Intakes lower than recommended were seen for Vitamin D, fiber, potassium, iron, Vitamin A, carbohydrates, and calcium. We found no large difference in adherence to DACH recommendations between men and women in CCSs, except for sodium (men: 236%, women: 149%), phosphorus (men: 187%, women: 150%), cholesterol (men: 125%, women: 103%), and iron (men: 97%, women: 56%) (Appendix A).

The mean AHEI score in CCSs was low at 49.8 (men: 47.7, women: 51.9) out of a maximum score of 110, and almost all individual components had a component score of less than 50% of the maximum score (Appendix A). Men’s scores were 57% for alcohol, 59% for fish, and 94% for trans-fat, while women scored 51% for polyunsaturated fatty acids (PUFA), 55% for sugar-sweetened beverage and fruit juice, 56% for red and processed meat, 57% for fish, and 94% for trans-fat. The comparison group scores were higher overall than those of CCSs: 53.7 for Bus Santé (51.8 for men and 54.4 for women) and 52.3 for CoLaus (50.2 for men and 54.0 for women). Figure 2 shows that the difference in score between CCSs and the general population was particularly large for vegetables, fruit, fish, and alcohol with lower scores for CCSs (all *p* < 0.001). Scores for sugar-sweetened beverage and fruit juice were higher in CCSs than in the general population (all *p* < 0.001).

### 3.3. Clinical Characteristics and Dietary Quality in CCSs

CCSs who received hematopoietic stem cell transplantation had a lower AHEI score compared to those not receiving transplantation (*p* = 0.029) (Table 2). We found no evidence that the dietary quality varied by type of cancer diagnosis (*p* = 0.071). Survivors of CNS tumors had an AHEI score of 47.2, whereas survivors of leukemia, lymphoma, and other cancers scored slightly higher. Among CCSs with CNS tumors, survivors of astrocytomas, the largest subgroup (41%), had an AHEI score of 46.1 (95% CI: 41.9, 50.3). We also found no evidence that the dietary quality was associated with time since diagnosis (*p* = 0.051). Age at diagnosis, and history of relapse were not associated with diet quality in CCSs. Cancer treatment, including cranial, chest, total body and/or abdominal radiation, glucocorticoids, anthracyclines, and alkylating agents were also not associated with diet quality.

## 4. Discussion

### 4.1. Principal Findings

We found that adherence to national dietary intake recommendations and diet quality among long-term CCSs was generally as poor as that of the general population in Switzerland. The diet of CCSs appears to be particularly low in fiber and too high in sodium and saturated fat, while reported alcohol intake was lower in CCSs than in the general adult population. In terms of diet quality, we observed a lower AHEI score in CCSs than in the general population. The AHEI scores for fish, fruit, vegetables, and alcohol were lower in CCSs than in the general population, whereas the score for sugar-sweetened beverages was higher. Diet quality at follow-up did not differ between cancer diagnosis, age and time since diagnosis, history of relapse, and cancer treatment exposures of CCSs. Except for hematopoietic stem cell transplanted CCSs who had a lower diet quality than those without. Our findings can inform the promotion of healthy eating in CCSs in Switzerland, as this population is at increased risk and earlier onset of chronic diseases.

### 4.2. Strengths and Limitations

For assessment of dietary intake and diet quality we used a validated FFQ, which may have led to misreporting of intake due social desirability and recall bias. The total reported calorie intake is rather low across groups, as an inherent limitation of the FFQ. Focus should be put on the relative difference between CCSs and comparison groups since the same FFQ was used across groups. Observed differences in dietary intake between CCSs and comparison groups could also have been due to seasonality. The SCCSS-Nutrition study distributed the FFQ, which asked about dietary intake of the previous four weeks, over a period of 6 months including spring and summer. Bus Santé and CoLaus collected data throughout the year. Differences in dietary intake could have further emerged among study populations because data on pregnancy and lactation were unavailable for the participants of the CoLaus and Bus Santé studies. However, since 44% of the women in the Bus Santé study were 40 years and older, and the CoLaus study included only women of 40 years and older the proportion of pregnant and lactating women that were included is expected to be relatively low. Finally, our data were based on a cross-sectional analysis and dietary intake of CCSs could change over time due to the occurrence of chronic health conditions or cancer recurrence [41,42]. 

Despite these limitations, this is the first study that directly compares dietary intake and quality of CCSs with adults from the general population in Switzerland. We included two comparison groups which assessed diet with the same FFQ as the CCSs. The SCCSS-Nutrition is strengthened by its national coverage, large sample size, and relatively high response rate, which makes our results representative for CCSs. Furthermore, we had access to high quality clinical information from the SCCR. 

### 4.3. Adherence to Dietary Guidelines and Diet Quality: Results in Relation to Other Studies

Our finding that the majority of CCSs does not fully adhere to national dietary recommendations is in line with previous studies among CCSs [6,10,14,15,41,42]. The low fiber intake we observed, of 12 g/day compared to the recommended DACH intake of ≥30 g/day [39], has also been reported for both short-term and long-term CCSs [6,11,43,44]. The St. Jude Lifetime Cohort Study found low intakes; among 1598 adult CCSs 26 years postdiagnosis [6] and 2570 adult CCSs 24 years postdiagnosis [11]. Smaller US studies with shorter follow-up times have found similar results [43]. The Chicago Healthy Living Study of 431 adult CCSs 19 years postdiagnosis found also a low fiber intake. This was lower than what was recommended by the 2010 Dietary Guidelines for Americans and even lower than the intake of the general population [44], which we also observed in our study. This finding is particularly salient for CCSs, who have a higher risk of developing chronic health conditions such as type 2 diabetes and CVD, and for whom high-fiber foods and whole grains are recommended to prevent these health problems [2,3]. Whether CCSs indeed adhere less to dietary recommendations than the general population will need to be confirmed in future studies. 

We observed excessive amounts of sodium, in line with other studies among CCSs [6,11,12,13,14,16]. CCSs may be especially burdened by excessive sodium intake, since they have an increased risk of developing high blood pressure and CVD due to earlier cancer treatment. In fact, we even found that high sodium intake was most marked among CCSs with CVD risk factors, and those who were overweight or obese [31], survivor groups who might particularly benefit from a healthier diet. We observed a lower alcohol consumption in CCSs compared to the general population, in line with our previous study [38]. This difference in alcohol consumption may be due to misreporting or intake variations between language regions [45] or age categories. Although the mean alcohol intake seems to be overall lower, CCSs in Switzerland seem to be more likely to drink alcohol frequently and to engage in binge drinking than the general population [46].

Our findings on low overall diet quality (45% of the maximum score for the modified AHEI) are in line with data from the St. Jude Lifetime Cohort of 2570 adult CCSs with a comparable time from diagnosis (24 ± 8 years) [11]. Similar results were observed in other studies from the US and Canada, where mean diet quality in CCSs ranged from 33–60% of the maximum score [8,12,13,14,47]. Although diet quality in these studies was measured with a different scoring method and is therefore not directly comparable with our results, they all found that diet quality was poor. It is important to look at the overall diet quality of CCSs as a high quality diet may prevent or delay the development of chronic health conditions, whereas a low quality diet may exacerbate this [10,38,47,48]. Recommendations for a healthy diet after treatment of childhood cancer emphasize on a sufficient consumption of fruit and vegetable, fiber and the limited consumption of fat, red meat, salt, and alcohol [48]. Therefore, it is even more striking that in our study the general population scored higher, particularly for fish, fruit, vegetables, and alcohol intake. This may reflect better eating behaviors within the general population compared to CCSs but may also be the result of a difference between language regions because CCSs came from German, French, and Italian speaking regions [45]. Results need to be compared and interpreted with caution, although we did not observe diet quality differences between language regions in CCSs (*p* = 0.642).

We found that CCSs who were treated with hematopoietic stem cell transplantation had lower diet quality scores than those without, but we did not find other differences in AHEI scores between subgroups of CCSs based on clinical characteristics. Compared to a larger study of the St. Jude Lifetime Cohort, we found a similar tendency that survivors of lymphoma had the highest AHEI score (51.2), followed by survivors of leukemia (50.2), whereas survivors of CNS tumors had the lowest score (47.2, *p* = 0.07) [11]. Lower diet quality scores among CNS tumor survivors may be the result of the exposure to several risk factors such as cranial radiation therapy (CRT) and surgical damage that can lead to impaired satiety signaling due to hypothalamic damage [48,49]. We did not find clear differences between AHEI scores when we looked at treatment exposures. This was consistent with previous literature on exposure with glucocorticoids, anthracyclines, and alkylating agents [11,13], but inconsistent for abdominal radiation. In the St. Jude Lifetime Cohort, survivors who received higher abdominal radiation doses had lower dietary quality scores compared to those who received lower doses [11]. We lacked power to stratify by dose and could not confirm these results. A smaller study including 91 CCSs showed that survivors exposed to CRT had lower HEI scores than those not exposed [13]. We found a similar trend, as did a US study among 22 acute lymphoblastic leukemia and lymphoma survivors [12]. No trend was seen for CRT stratified by dose in the St. Jude Lifetime Cohort among 916 CCSs who received CRT [11]. Since cancer diagnosis directly affects the type of treatment a patient receives, occurrence of late effects can play an important role in the survivors’ dietary eating habits. Large-scale prospective studies with a homogenous study population and repeated dietary assessments are therefore needed to better investigate diet quality between patients exposed to different types of treatment.

### 4.4. Implications and Recommendations 

Dietary recommendations for childhood and adult cancer survivors emphasize the importance of a balanced diet with a high intake of fruit, vegetable, and fiber, a limited intake of alcohol, and a low intake of red and processed meat to reduce the development of nutrition-related chronic diseases after cancer treatment [10,36,50]. Despite these dietary recommendations for cancer survivors, we found low AHEI scores in CCSs, even lower than in the general population. This confirms our previous findings on fish and vegetable consumption [38]. However, it is unclear to what extent CCSs are aware of these dietary recommendations, whether they are communicated to them by health care professionals, and whether CCSs perceive diet as a risk factor for late effects. However, given the strong evidence concerning diet and health in general and the increasing data for CCSs [5], more focus should be placed on the importance of balanced eating habits and physical activity during annual long-term follow-up visits as CCSs could reduce their risk of late effects through behavioral changes. 

## 5. Conclusions

We found a poor dietary intake and diet quality in CCSs in Switzerland that was as poor as that reported in the general adult population. Low fruit, vegetable and fiber intake and the high intake of sugar-sweetened beverages, red and processed meat, sodium, and saturated fat are of particular concern. Our findings suggest that previous cancer treatment exposure and cancer characteristics such as age at and time since diagnosis do not substantially influence the diet quality of CCSs in the long term. We suggest population-based interventions reinforcing the importance of a healthy diet to everyone.

## Figures and Tables

**Figure 1 nutrients-13-01767-f001:**
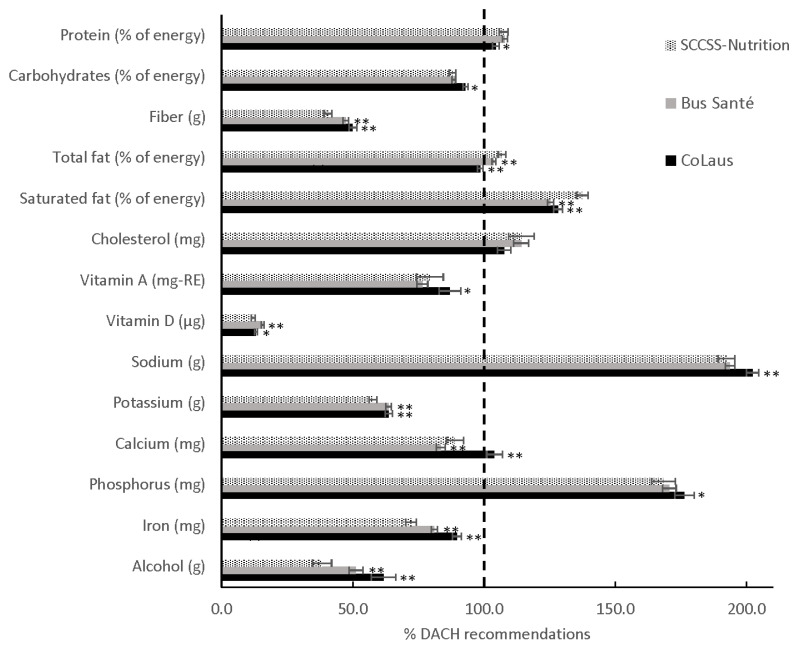
Nutrient intake compared with DACH recommended intake or limit in childhood cancer survivors (SCCSS-Nutrition) and the general population in Switzerland (Bus Santé and CoLaus) ^a^. The length of the bar per nutrient corresponds to the percentage of mean intake (95% CIs) compared to the recommended intake level * 100. Recommended intake is estimated on the basis of age and sex according to dietary recommendations for Germany (D), Austria (A) and Switzerland (CH) (DACH) 2015, weighted by the age and sex distribution per study population. For alcohol intake the maximum tolerated dosage was taken. Nutritional goals were set at 100 when the mean intake met the recommended intake or the maximum tolerated dosage. ^a^ Standardized on sex and age at survey according to CCSs. * *p* < 0.05 compared to CCSs. ** *p* < 0.001 compared to CCSs.

**Figure 2 nutrients-13-01767-f002:**
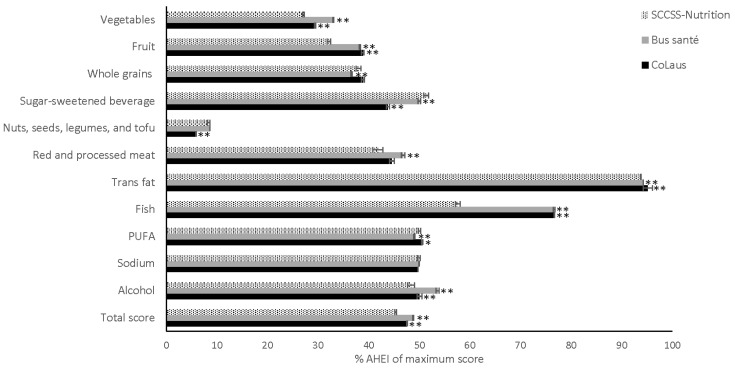
Percentages of Alternate Healthy Eating Index ^a,b^ of maximum score with 95% CIs in childhood cancer survivors (SCCSS-Nutrition) and the general population in Switzerland (Bus Santé and CoLaus) ^c a^ Adapted from Chiuve et al. [40] ^b^ Intermediate food intake was scored proportionately between the minimum score zero and the maximum score ten. The maximal total AHEI score was 110. ^c^ Adjusted for sex, age at survey, and if applicable ICCC-3 diagnosis, general population is additionally standardized on sex and age at survey according to CCSs. * *p* < 0.05 compared to CCSs. ** *p* < 0.001 compared to CCSs.

**Table 1 nutrients-13-01767-t001:** Characteristics of childhood cancer survivors and the general population in Switzerland.

Characteristics	CCSs (*n* = 775)	Population-Based Samples			
		Bus Santé ^a^ (*n* = 2529)		CoLaus ^a^ (*n* = 1276)	
		*n* (%_std_)	*p* ^b^	*n* (%_std_)	*p* ^b^
Sex, *n* (%)					
Men	386 (50)	1229 (50)	NA	628 (59)	NA
Age at survey, *n* (%)					
20–24 yrs	69 (9)	246 (16)	NA	-	NA
25–29 yrs	179 (23)	298 (16)		-	
30–34 yrs	164 (21)	391 (18)		-	
35–39 yrs	156 (20)	475 (18)		-	
40–44 yrs	123 (16)	499 (16)		552 (61)	
45–50 yrs	84 (11)	620 (16)		724 (39)	
Language region, *n* (%)					
German speaking	555 (72)	-	NA	-	NA
French speaking	202 (26)	2529 (100)		1276 (100)	
Italian speaking	18 (2)	-		-	
Country of birth, *n* (%)					
Switzerland	738 (95)	1167 (51)	<0.001	750 (59)	<0.001
Other	37 (5)	1362 (49)		526 (41)	
Education (highest degree)*, n (%)*					
Lower than university	537 (69)	1275 (52)	<0.001	915 (70)	0.149
University	238 (31)	1242 (48)		360 (30)	
Missing	-	12 (<1)		1 (<1)	
Living situation, *n* (%)					
Alone	156 (20)	997 (45)	<0.001	237 (20)	0.621
Other	614 (79)	1531 (55)		1036 (80)	
Missing	5 (<1)	1 (<1)		3 (<1)	
Physical activity ^c^*, n (%)*					
Inactive	159 (21)	314 (27)	0.002	295 (22)	<0.001
Active	614 (79)	904 (72)		877 (70)	
Missing	2 (<1)	8 (<1)		104 (8)	
Smoking status, *n* (%)					
Never	515 (66)	1302 (53)	<0.001	566 (45)	<0.001
Former	129 (17)	609 (22)		398 (30)	
Current	122 (16)	617 (25)		311 (24)	
Missing	9 (1)	1 (<1)		1 (<1)	
BMI at survey, *n* (%)					
Underweight	40 (5) ^d^	78 (4)	0.015	19 (1)	<0.001
Normal	488 (63)	1580 (64)		643 (48)	
Overweight	170 (22)	641 (24)		451 (38)	
Obese	74 (10)	199 (7)		157 (12)	
Missing	3 (<1)	31 (1)		6 (<1)	

Abbreviations: CCS, childhood cancer survivors; NA, not applicable; std, standardized. ^a^ Standardized on sex and age at survey according to CCSs. ^b^
*p* value calculated from chi-square statistics comparing comparison groups with CCSs (2-sided test). ^c^ CCSs: < or ≥150 min of moderate, or 75 min of vigorous physical activity or a combination of moderate and vigorous activity per week. Bus Santé and CoLaus: < or ≥ 1st quartile of total weekly physical activity time excluding sleep. The survey years 2017 (*n* = 689) and 2018 (*n* = 615) in Bus Santé have no physical activity information. Bus Santé results are only shown for survey year 2015 (*n* = 642) and 2016 (*n* = 585). ^d^ Self-reported BMI in CCSs.

**Table 2 nutrients-13-01767-t002:** Diet quality in childhood cancer survivors by clinical characteristics (retrieved from ANCOVA) ^a^.

Characteristics	CCSs (*n* = 775)		
	*n* (%)	AHEI Score (95% CI)	*p*
ICCC-3 diagnosis			
I: Leukemia	238 (31)	50.2 (48.7, 51.6)	0.071
II: Lymphoma	163 (21)	51.2 (49.4, 53.0)	
III: CNS tumor	79 (10)	47.2 (44.7, 49.7)	
Other	295 (38)	49.5 (48.2, 50.8)	
Age at diagnosis, yrs			
<5	251 (32)	48.7 (47.1, 50.3)	0.304
5–9	163 (21)	50.5 (48.7, 52.3)	
10–14	225 (29)	49.8 (48.2, 51.3)	
15–20	136 (18)	51.2 (49.0, 53.3)	
Time since diagnosis, yrs			
≤25	367 (47)	51.0 (49.6, 52.5)	0.051
>25	408 (53)	48.7 (47.4, 50.1)	
History of relapse			
No	686 (89)	49.9 (49.1, 50.8)	0.431
Yes	89 (11)	48.9 (46.5, 51.3)	
Treatment exposures			
Radiation			
Cranial			
No	655 (85)	50.0 (49.2, 50.9)	0.240
Yes	120 (15)	48.6 (46.5, 50.8)	
Chest			
No	686 (89)	49.8 (49.0, 50.7)	0.909
Yes	89 (11)	49.7 (47.1, 52.3)	
TBI and/or abdominal			
No	707 (91)	49.9 (49.0, 50.7)	0.631
Yes	68 (9)	49.2 (46.4, 52.0)	
Glucocorticoids			
No	444 (57)	49.3 (47.9, 50.7)	0.401
Yes	331 (43)	50.5 (48.7, 52.2)	
Anthracyclines			
No	482 (62)	49.5 (48.4, 50.5)	0.342
Yes	293 (38)	50.4 (49.0, 51.8)	
Alkylating agents			
No	454 (59)	49.3 (48.2, 50.4)	0.167
Yes	321 (41)	50.5 (49.2, 51.8)	
Hematopoietic stem cell transplantation			
No	745 (96)	50.0 (49.2, 50.8)	0.029
Yes	30 (4)	45.3 (41.2, 49.4)	

Abbreviations: CNS, central nervous system; CCS, childhood cancer survivors; ICCC-3, International Classification of Childhood Cancer, Third Edition; TBI, total body irradiation. ^a^ Adjusted for sex, age at survey, and language region. Treatment characteristics were additionally adjusted for ICCC-3 diagnosis.

## Data Availability

The data presented in this study are available on request from the corresponding author.

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
