# Peer review of "Dietary Intake and Diet Quality of Adult Survivors of Childhood Cancer and the General Population: Results from the SCCSS-Nutrition Study"

_nutrients, 2021, doi:10.3390/nu13061767_

Round 1

Reviewer 1 Report

Thank you for such a comprehensive research paper. There has been a growing literature looking at the diet quality of CCS but most studies have lacked a control group and have been a small sample size. It is great to see a paper addressing both issues. Overall, this is a well-written paper and I only have a few suggestions to make this paper stronger. My main comment is that with so much information, the paper is hard to follow at times. My other main comment is that there is now much more of a focus on diet variety over nutrient intake in research as “we don’t eat nutrients we eat food”. In this paper you have both types of information, which is important, but I feel you have highlighted the nutrient aspects of the paper and the diet quality information gets a bit lost. Suggest looking at reducing the focus on nutrient information and highlight the diet quality more.

Abstract

Re-write abstract with not so many numbers for the nutrient intake section-suggest highlighting the important nutrients. I also think that you need to highlight the results from the individual diet quality scores for such nutrients as fruit/vegetables rather than focusing on the nutrients. The type of information in the principles finding section in the discussion is more the kind of data I suggest highlighting in the abstract

Methods

Further justification of dates of data collection of control group versus CCS group as they are all collected at different years especially the Bus Sante which doesn’t overlap with the CCS data collection. Please justify why this difference is acceptable

Results

In the section “Dietary intake and diet quality in CCSs and comparison groups”  this section is hard to follow with so many percentages and much of the data is in table 1. Suggest finding a way to incorporate into the figure or present another way.. Figure 1 is easy to read.

Table 1 formatting needs improving. It is hard to read especially in column 1 with the words going over two lines and the word “vegetable” is spelt wrong.

Figure 2 consider adding in a notation to show which values were significantly different.

In section 3.3. Clinical characteristics and dietary quality in CCSs 283

“CCSs who received hematopoietic stem cell transplantation had a lower AHEI score 284 compared to the others (P = 0.029)” Who do you mean by others?

“There was weak evidence that the dietary quality varied by type of cancer diagnosis (P = 0.071) & We also found 289 weak evidence that the dietary quality was associated with time since diagnosis (P = 290 0.051)”, .-avoid using the words “weak evidence”. P is >0.05 therefore your conclusion should be that there is no statistically significant difference

Discussion

Section 4.1: Not sure of use of the word “worse” in the line “The AHEI scores for fish, fruit, vegetables, and alcohol were 302 worse in CCSs than in the general population, whereas the score for sugar-sweetened beverages was better.”  suggest re-wording.

Section 4.1 Be specific about clinical characteristics in the line “Diet quality at follow-up did not differ between clinical characteristics of CCSs”

4.3. Adherence to dietary guidelines and diet quality: results in relation to other studies- way too many numbers in this section on fibre so hard to follow. In the discussion it is OK to be more general and then readers can read the specifics of the intakes themselves

“They found a mean ± SD Healthy Eating 362 Index 2010 score (HEI-2010) of 57.9 ± 12.4 out of a maximum score of 100 [10” This is already mentioned in the intro, no need to repeat, keep it more general in the discussion.

 In section 4.5 you state: “Dietary recommendations for childhood and adult cancer survivors emphasize the importance of a balanced diet with a high intake of fruit, vegetable, and fiber and a low intake of alcohol and red and processed meat to reduce the development of nutrition- related chronic diseases after cancer treatment” I feel like this is an important discussion point and needs to be made earlier in the discussion. Your opening sentence in the paper is talking about a healthy diet and how this is a modifiable risk factor but there is little mention of how the results of the diet quality section related to healthy diets and healthy dietary patterns in CCS. Suggest unpacking this more in the discussion rather than mentioning it only in the recommendations.

“The importance of a healthy lifestyle was also confirmed in a study of 1598 CCSs 26 years postdiagnosis [6], which found that an un-410 healthy lifestyle based on anthropometric, FFQ, and physical activity data according to World Cancer Research Fund/American Institute for Cancer Research recommendations…” I think this is more suited to your introduction rather than recommendations. Suggest not bringing in new literature in your recommendations section.

Reviewer 2 Report

The manuscript entitled “Dietary intake and diet quality of adult survivors of childhood cancer and the general population: Results from the SCCSS-Nutrition study” presents interesting issue, but some areas must be corrected.

Major:

  1. In spite of the fact that Authors included in their Materials and methods Section the sub-chapter entitled Statistical Analysis, in fact they did not conduct any statistical analysis of their results. The results presented in Table 2, Figure 1 and Figure 2 (the results of the nutritional assessment) are not analyzed at all, while they are the most important results of the conducted study. Authors analyzed only the general characteristics of the studied group (Table 1). Authors should at least compare the share of sub-groups meeting the recommendation of intake of specific nutrients (chi-2), but the deepen analysis are also highly recommended. Without any statistical analysis, it is impossible to conclude based on the conducted description only.
  2. Authors should not reproduce their data in various forms – in the current version of the paper, the same data tar presented in Table 2, Figure 1 and in the text (3 times the same information).
  3. The reference groups of Bus Santé and CoLaus studies seem to be totally incomparable for the studied group, as they differ while compared with the studied group, taking into account age, gender and ethnicity. Taking this into account, it is hard to conclude about the representativeness of the studied group.
  4. There are some ethical doubts, as Authors indicated the ethical committee agreement for SCCSS-Nutrition study, but in fact they used not only data from SCCSS-Nutrition study, but also from Bus Santé and CoLaus studies. For the indicated studies, Authors did not provide the confirmation that they obtained ethical committee agreement to use them.
  5. Authors should properly formulate their statements to be in agreement with general guidelines – e.g. Authors indicated that ‘To prevent these health problems, it is recommended […] to consume alcohol in moderation’ – Authors suggest that if one does not consume alcohol, he should include it to a diet in moderation.

Abstract:

Authors should properly formulate the aim of their study (e.g. “The aim of the study was…”) instead of indicating only what was done.

Authors should provide results of statistical analysis.

The conclusions must be based on the results of a statistical analysis.

Introduction:

Authors should properly formulate their statements to be in agreement with general guidelines – see above.

Authors should deepen the presented justification of their study, while presenting the information from other countries.

Authors should properly formulate the aim of their study (e.g. “The aim of the study was…”) instead of indicating only what was done.

Materials and Methods:

Authors should properly justify using Bus Santé and CoLaus studies as a reference, as they seem to be totally incomparable for the studied group, as they differ while compared with the studied group, taking into account age, gender and ethnicity – see above

Authors should provide detailed information about applied FFQ (its validation, questions, food items, etc), as well as methodology of the study (how was it conducted).

Authors should provide information about statistical analysis which should be conducted.

Results:

Authors should include statistical analysis  - see above

Authors should not reproduce data in tables, figures and in the text – see above

Figure 2 should be replaced by table to be easier to follow

Discussion:

It is not justified to formulate recommendations based on the conducted study, due to really important sources of bias (see above - representativeness)

Authors should in their discussion include 3 areas: (1) compare gathered data with the results by other authors, (2) formulate implications of the results of their study and studies by other authors, (3) formulate the future areas which should be studied.

Each statement which is formulated should be based on the conducted statistical analysis

Conclusions:

Each statement which is formulated should be based on the conducted statistical analysis

References:

Authors should include adequate references, while self-citations should be avoided, as they are not adequate (14 self-citations is an excessive amount).

Round 2

Reviewer 2 Report

The manuscript entitled “Dietary intake and diet quality of adult survivors of childhood cancer and the general population: Results from the SCCSS-Nutrition study” presents interesting issue, but some areas must be corrected.

Major:

The reference groups of Bus Santé and CoLaus studies seem to be totally incomparable for the studied group, as they differ while compared with the studied group, taking into account age, gender and ethnicity. We may see it especially for the age in case of CoLaus, as for the studied group age is 20-50, while for CoLaus, it is 40-60. Taking this into account, it is hard to conclude about the representativeness of the studied group and the comparison (especially with CoLaus) seems to be not justified.

Abstract:

The conclusions must be based on the results of a statistical analysis. If he AHEI scores for fish, fruit, vegetables, and alcohol were worse in CCSs than in the general population, whereas the score for sugar-sweetened beverages was better, Authors cannot state that “long-term CCSs and the general population have similar poor dietary intake and quality in Switzerland” but they should reflect differences.

Introduction:

Authors should deepen the presented justification of their study, while presenting the information from other countries.

Materials and Methods:

Authors should properly justify using Bus Santé and CoLaus studies as a reference, as they seem to be totally incomparable for the studied group, as they differ while compared with the studied group, taking into account age, gender and ethnicity – see above

Authors should provide detailed information about applied FFQ: its validation (what were the results of validation), questions (how were they formulated), food items (which groups were included), etc, as well as methodology of the study (how was it conducted). As the FFQ which was applied is the crucial element of the study, it should be presented with all necessary details.

Results:

Figures should be replaced by tables to be easier to follow

Discussion:

It is not justified to formulate recommendations based on the conducted study, due to really important sources of bias (see above - representativeness)

Authors should in their discussion include 3 areas: (1) compare gathered data with the results by other authors, (2) formulate implications of the results of their study and studies by other authors, (3) formulate the future areas which should be studied.

Each statement which is formulated should be based on the conducted statistical analysis

Conclusions:

Each statement which is formulated should be based on the conducted statistical analysis

References:

Authors should include adequate references, while self-citations should be avoided, as they are not adequate (9 ones - 20% of self-citations in the manuscript is an excessive amount).

Round 3

Reviewer 2 Report

The manuscript entitled “Dietary intake and diet quality of adult survivors of childhood cancer and the general population: Results from the SCCSS-Nutrition study” presents interesting issue, but some areas must be corrected. Unfortunately, Authors either deliberately ignore my comments, or they do not understand them.

Major:

The reference groups of Bus Santé and CoLaus studies seem to be totally incomparable for the studied group, as they differ while compared with the studied group, taking into account age, gender and ethnicity. Authors seem to not understand, that comparing the studied group (aged 20-50 years) with a group older than 40 is not a proper approach (40-years old individual may not be a reference for 20-years old one)

Taking this into account, it is hard to conclude about the representativeness of the studied group and the comparison (especially with CoLaus) seems to be not justified.

Introduction:

Authors should deepen the presented justification of their study, while presenting the information from other countries.

Materials and Methods:

Authors should properly justify using Bus Santé and CoLaus studies as a reference, as they seem to be totally incomparable for the studied group, as they differ while compared with the studied group, taking into account age, gender and ethnicity – see above

Authors should provide detailed information about applied FFQ: its validation (what were the results of validation), questions (how were they formulated), food items (which groups were included), etc, as well as methodology of the study (how was it conducted). As the FFQ which was applied is the crucial element of the study, it should be presented with all necessary details.

Results:

Figures should be replaced by tables to be easier to follow

Discussion:

It is not justified to formulate recommendations based on the conducted study, due to really important sources of bias (see above - representativeness)

Authors should in their discussion include 3 areas: (1) compare gathered data with the results by other authors, (2) formulate implications of the results of their study and studies by other authors, (3) formulate the future areas which should be studied.

Each statement which is formulated should be based on the conducted statistical analysis

Conclusions:

Each statement which is formulated should be based on the conducted statistical analysis (see above - representativeness)

References:

Authors should include adequate references, while self-citations should be avoided, as they are not adequate (9 ones - 20% of self-citations in the manuscript is an excessive amount).

Author Response

This manuscript is a resubmission of an earlier submission. The following is a list of the peer review reports and author responses from that submission.